# Technical Study of Automated High-Throughput High-Sensitive Ceruloplasmin Assay on Dried Blood Spots—Reinstate the Potential Use for Newborn Screening of Wilson Disease

**DOI:** 10.3390/ijns8040064

**Published:** 2022-12-01

**Authors:** Chloe Miu Mak, Ching Tung Choi, Tsz Ki Wong, Hanson Heearn Chin, Hillman Kai Yin Lai, Koon Yu Yuet

**Affiliations:** Newborn Screening Laboratory for Inborn Errors of Metabolism, Department of Pathology, Hong Kong Children’s Hospital, Hong Kong SAR, China

**Keywords:** newborn screening, Wilson disease, ceruloplasmin, high-sensitive assay, dried blood spot

## Abstract

In this study, we modified a fully automatic immunoassay on ceruloplasmin concentration on dried blood spots (DBS) to increase its analytical sensitivity in order to accurately differentiate newborns from true Wilson disease (WD) patients. Modifications to the assay parameters of the Roche/Hitachi Cobas c systems immunoturbidimetric assay are adjusted to lower the limit of quantitation to 0.60 mg/L from 30 mg/L. This enables sensitive measurement of ceruloplasmin in eluent after DBS extraction. In addition, reference intervals and receiver operating characteristic curve analysis for diagnostic cut-off were established using DBS of neonates and WD adult patients. After DBS whole blood calibration, the 95th percentile of the reference interval for newborns was 86–229 mg/L. The cut-off value of 54 mg/L was found to be the most optimal point for differentiating true adult WD from newborn controls. This test shows a high area under curve of 1.000 with 100% sensitivity and specificity in differentiating normal newborns from WD adult samples. However, the results should be further validated with true newborn WD patient samples together with the consideration of other factors that can also lead to low ceruloplasmin levels. This test shows application potential in newborn screening for WD, which can save lives through early identification and timely treatment.

## 1. Introduction

Wilson disease (WD, MIM #277900) is a potentially fatal disorder of systemic copper accumulation and intoxication with a presentation of neurological deficits, psychiatric illnesses, and liver failure. The age of onset ranges from infancy to adulthood. WD is a progressive and invariably fatal disorder if left untreated. Early and accurate diagnosis of presymptomatic patients is critical to prevent irreversible liver or brain damage. De-coppering agents, e.g., penicillamine or trientine, provide cheap and effective treatment. Early diagnosis and treatment result in a good prognosis and thus avoid unnecessary morbidities and mortalities. This is especially true for presymptomatic pediatric patients starting with low copper body loads. They can be effectively treated with zinc, which is much safer than penicillamine [1]. These patients can enjoy normal life quality and expectancy. In contrast, without timely treatment, the disease leads to irreversible liver damage requiring a transplant, neurological impairment, and even death.

The global genetic prevalence of WD is reported as 1 in 7149 [2], while WD is more prevalent among Asians, e.g., 1 in 5400 in Hong Kong [3]; 1 in 3667, in Korea [4]; 1 in 1395 [5] to 1 in 8055 [6] in Japan. It is one of the most common hereditary liver diseases. Without newborn screening (NBS), under- or misdiagnosis of WD is common as a result of its protean presentations, low clinical suspicion, and limited diagnostic test performance.

Conventionally, WD is diagnosed by low serum ceruloplasmin, the presence of Kayser-Fleischer rings, and typical neurological deficits. A more comprehensive scoring system was developed, first by the Working Group at the 8th International Meeting on Wilson’s disease in Leipzig in 2001 [7] and further incorporated into the European Association for the Study of the Liver (EASL) Clinical Practice Guidelines on WD in 2012 [8]. Other major markers include daily urine copper excretion (baseline and penicillamine challenge), a relative exchangeable serum copper index, the presence of Coombs-negative hemolytic anemia, a hepatic copper level, and genetic tests with variable sensitivities. In addition, the final diagnostic accuracy depends largely on how clinically florid the patient presents and how extensive investigations are conducted.

Although WD is considered a suitable target for NBS, currently the possibility of NBS is hindered by the lack of a sensitive assay with a cost-effective capacity for mass screening. Ceruloplasmin activity has been shown to be both sensitive and specific for the detection of WD. However, the assays are laborious, and so far, there is no commercial kit available, while a high-throughput assay of ceruloplasmin concentration measurement is enabled by an automated immunoassay platform. However, the commercial assays are designed for clinical diagnosis at a time when WD patients present at older ages, and the detection limit is not sensitive enough to accurately differentiate normal newborns from a true WD, whose levels are much lower than those of older children and adults.

In this study, we modified a fully automatic immunoassay on ceruloplasmin concentration on dried blood spots (DBS) to increase its analytical sensitivity in order to accurately differentiate newborns from true WD adult patients and investigate the technical feasibility of its use in NBS for WD. In addition, reference intervals for newborns on DBS ceruloplasmin and receiver operating characteristic (ROC) curve analysis for the diagnostic cut-off for NBS of WD were established.

## 2. Materials and Methods

### 2.1. In-House Modification of the ROCHE Ceruloplasmin Assay

In vitro immunoturbidimetric assay for the quantitative determination of ceruloplasmin in human serum and plasma on Roche/Hitachi cobas c502 system (reference number 20764663, Roche Diagnostics GmbH, Mannheim, Germany) was applied on DBS extracts. Roche Cobas c502 is a medium throughput automated clinical chemistry module that performs photometric assay tests using test tubes loading with up to 600 tests/h. Instrument settings included endpoint assay with sample blank, reaction time 10 min, photometric measurement points at 83.3 s and 308.6 s, primary wavelength 340 nm, secondary wavelength 700 nm, sample volume 20 µL with 5-fold dilution (original 11 µL with 11-fold dilution), reagent volume R1 100 µL and R2 20 µL. R1 contained accelerator, polyethylene glycol (50 g/L) in phosphate buffer and R2 contained anti-ceruloplasmin T antiserum (rabbit) specific for human ceruloplasmin in phosphate buffer. Roche calibrator for automated systems prealbumin-ASLO-Ceruloplasmin (c.f.a.s. PAC, reference number 03555941) pre-diluted with extraction phosphate buffer in 1:1 as stock calibrator, and the analyzer pre-diluted the stock calibrator automatically for 6-point calibration (0, 1.37, 1.83, 2.74, 4.60, and 9.14 mg/L) to extend the analytical range. DBS calibrators were prepared by Ceruloplasmin, Human Plasma (CAS 9031-37-2, Sigma-Aldrich, Saint Louis, USA) diluted to different concentrations (0, 10, 20, 40, 100, 200, and 400 mg/L) using phosphate buffered saline (pH 7.4) with bovine serum albumin.

### 2.2. Measurement of Ceruloplasmin in DBS

Our laboratory provides a DBS metabolic screening service for patients suspected of amino acids, organic acids, and fatty acid oxidation disorders. Residual DBS samples of 350 patients (full term, aged 0–28 days) without clinical suspicion of WD from 1 October 2018 to 31 December 2020 stored at −20 °C with desiccators were retrieved for the analysis. The residual DBS were anonymous and de-identified. In addition, 7 adult patients with known WD and 2 heterozygotes from the Hong Kong Wilson Disease Association were invited for this study for DBS collection. Informed consent was obtained from the adult patients. DBS samples were warmed at room temperature before test and three discs of 3.2 mm diameter were punched into microtiter plate with addition of 150 µL phosphate-buffered saline buffer. The puncher used was Puncher^®^ Instrument (part no. 1296-071, PerkinElmer, Inc., Waltham, MA, USA). The samples were vortexed briefly and spun down to make sure the disc was soaped within the buffer and shaken for 4 h. After overnight incubation at room temperature, the samples were centrifuged at 10 min × 3000 g and subjected to analysis. Reference intervals and ROC curve analysis were performed using MedCalc (Version 20.027, MedCalc Software Ltd., Ostend, Belgium).

## 3. Results

### 3.1. Performance of the Modified ROCHE Ceruloplasmin Assay

The manufacturer-claimed limit of detection for the Roche/Hitachi Cobas c systems immunoturbidimetric assay for plasma/serum/eluent was 30 mg/L. This is not adequate for neonates and most WD patients with much lower ceruloplasmin concentrations. Moreover, because of the incomplete recovery after DBS extraction, the ceruloplasmin levels in the DBS eluent, before DBS calibration, were at least one order of magnitude lower (ranged 1.16–6.18 mg/L for DBS eluent control; 0.90–1.15 mg/L for WD samples). It would hence be difficult to detect and differentiate presymptomatic WD patients from normal neonates without a sensitive quantification. Three levels of diluted Liquichek Immunology Control (Cat #68991, Bio-Rad Laboratories, Hercules, CA, USA) were assayed with 20 replicates on three independent days. Results were CV 7.7% at a mean of 0.90 mg/L; 11.8% at 0.69 mg/L; 11.6% at 0.37 mg/L. Hence, the limit of quantitation of 0.60 mg/L was adopted. Three levels of the same control materials were assayed with 25 replicates over five independent days for between-batch precision. Results were CV 4.1% at a mean of 0.76 mg/L; 2.4% at 1.54 mg/L; 0.6% at 7.27 mg/L. In addition, no significant changes in ceruloplasmin levels were observed in DBS samples stored at −20 °C under desiccants over two years.

### 3.2. Reference Intervals Established for Newborn on DBS Ceruloplasmin

The mean ± SD ceruloplasmin concentration in DBS was 141 ± 37.5 mg/L (after DBS calibration). The 95th percentile of the reference interval for newborns aged 0–28 days based on the non-parametric percentile method (CLSI C28-A3) was 86–229 mg/L for DBS.

### 3.3. ROC Analysis of for Diagnostic Cut-Off

A cut-off value of 54 mg/L for ceruloplasmin level in newborn DBS was considered to be the most optimal point for differentiating true adult WD cases from newborn controls in our locality. This screening test shows a high area under curve of 1.000, with 100% sensitivity and specificity (Figure 1) in differentiating normal newborns from adult WD samples.

## 4. Discussion

WD has always been an attractive target for NBS. It fulfills Wilson and Jungner’s 1968 screening criteria, except with no cost-effective, specific biomarkers or screening methods. In previously published methods such as sandwich enzyme-linked immunosorbent assay of holoceruloplasmin [9] and peptide immune-SRM assay of ATP7B protein [10], it is usually laborious that is not suitable for high-throughput screening. Hence, our study proposes to modify an automated immunoassay such that it can be fitted for screening purposes.

We successfully lowered the analytical sensitivity two orders of magnitude below the current assay from 30 mg/L to 0.6 mg/L to meet the lowered ceruloplasmin level in DBS eluent. After DBS calibration, we established the 95th percentile reference interval for babies aged 0–28 days as 86–229 mg/L together with the diagnostic cut-off of 54 mg/L. The reference interval is comparable with CALIPER serum ceruloplasmim, which 74–237 mg/L at ages 0 to <2 months old (https://caliperdatabase.org/#/, accessed on 16 September 2020). At this cut-off, the test shows a high area under curve of 1.000, with 100% sensitivity and specificity in differentiating normal newborns from WD adult samples. However, the results should be further validated with true WD newborn samples, together with the consideration of other factors that can also lead to low ceruloplasmin levels. The high-throughput Roche cobas 501 instrument or other similar models commonly available in routine chemistry laboratories can be adopted for this modified assay. The reagent cost for one ceruloplasmin test is about USD 1. In addition, the process is automated and sample preparation is simple, enabling a high throughput for mass screening.

Despite many attempts at research trials to determine the optimal screening age for WD, the outcome remains disputable. In Yamaguchi’s study, 3 years old is suggested to be the best sampling age for WD screening [6], which is then cited by subsequent studies, such as Nakayama [11]. Although Yamaguchi’s study was the biggest study so far, several limitations and pitfalls were observed in the study design. Firstly, no WD patients were identified among 126, 820 newborns, whereas there should have been 3–4 patients according to the previously published incidence rates in Japan. There could be several reasons explaining why no WD patients were detected. First, different cut-off values were used by the 10 hospitals in the study, ranging from 15 to 100 mg/L, and this may cause false negatives. Kroll et al. reported two WD patients with their NBS DBS ceruloplasmin at 26 and 28 mg/L, respectively [12]. So, some true cases might be interpreted as normal in some hospitals using a lower cut-off. Compared with the screening trial for WD from late infancy to elementary school in the same Yamaguchi’s study, a standardized and higher cutoff of 100 mg/L was applied to all 24,165 cases, with five WD cases detected. Second, there were significant numbers of screen-positive newborns who defaulted in the first (23.7%) and second (22.4%) re-examinations in Yamaguchi’s study. While they belonged to the more borderline cases, they were more likely to be true cases and such default may result in the missing of true cases. In contrast, in the screening trial for WD from late infancy to elementary school, there was no default. Therefore, the conclusion from Yamaguchi’s study that the neonatal stage is not an optimal fit for WD screening is still questionable. Table 1 shows the comparison between different studies on WD screening using ceruloplasmin. Apart from Yamaguchi’s study, the other studies, including our study, show supportive evidences for the use of DBS ceruloplasmin as a potential marker for NBS for WD [4,5,6,12].

It is well known that the age of onset of WD varies from infancy to adulthood and pathological findings can be detectable as early as 1 year old [13]. Therefore, screening should be performed as soon as possible when the body copper load is low to prevent these pathological damages, and the earlier the treatment is initiated, the better the outcome is. Conceivably, if we are able to identify WD patients at a neonatal stage when their body copper load is low in the presymptomatic stage, we would be able to administer zinc therapy timely enough to control the disease trajectory. It should significantly improve the clinical outcome of these patients and enable them to maintain a normal life.

Regarding our limitations, first, we were not successful in retrieving WD newborn DBS because the DBS-based NBS program was only started in 2015 in Hong Kong. However, we managed to identify that the adult WD DBS ceruloplasmin levels are still significantly lower than those of normal newborns. Since WD is a genetic disease, the levels of ceruloplasmin of WD in the neonatal period and adulthood should be comparable. In WD, the defective ATP7B protein fails to incorporate copper into apoceruloplasmin, which has a very short half-life and degrades rapidly after being secreted into the circulation, leading to a low or undetectable blood ceruloplasmin level in WD. It is conceivable that the blood ceruloplasmin levels in WD patients at newborn and adult age are comparable. Further studies with the inclusion of true newborn WD patient samples are required. Second, the sample size of WD patients recruited for this study is small. In order to improve the accuracy of the cut-off and minimize false negative results, more patient samples will be needed to refine the cut-off, ideally from newborn patients. Third, preterm babies were not included in this study. Their ceruloplasmin levels are expected to be lower than those of full-term babies in view of their immature liver function. Fourth, since ceruloplasmin is an acute-phase protein, its level increases during acute stresses and might give false negative results. In situations with preterm babies and babies suffering from acute stresses, we recommend following the CLSI protocol (NBS03) that a second DBS sample should be taken again upon discharge or on day 28 of life (whichever comes first). Such first results can be reported as “incomplete” without recall, similar to the logistics in some NBS programs for severe combined immunodeficiency. Low ceruloplasmin results in full-term babies as first-tier screening could be followed by second-tier genetic testing to reduce false positives. Fifth, about 10% of heterozygotes and 10% of WD patients will have low and normal ceruloplasmin levels, respectively. Although it is not the most ideal marker, the false negative rate however is comparable with NBS for congenital adrenal hyperplasia and citrin deficiency using markers of 17-hydroxyprogesterone and citrulline, respectively, for which NBS programs are still successful.

## 5. Conclusions

In conclusion, we successfully modified the automatic immunoassay for ceruloplasmin concentration on DBS. Its analytical sensitivity is lowered by two orders of magnitude from the manufacturer’s claim and enhanced to differentiate normal newborns from true WD patients with almost undetectable ceruloplasmin levels. Since the process is automated and high throughput is allowed, it is feasible to add this test to the current DBS-based NBS program. A reference interval and cut-off were established, which are subjected to further refinement when more data can be collected. The NBS for WD enables early identification of WD patients and therefore timely treatment to prevent devastating damage and improve WD patients’ quality of life and expectancy. In addition, other diseases, such as hereditary aceruloplasminemia and Menkes disease, might also be identified during the screening and added to the coverage of screening targets, which further widens the scope of genetic diseases we can screen for. In the near future, when the cost of next-generation sequencing is getting less and less prohibitive, first-tier next-generation sequencing targeted for *ATP7B* pathogenic and likely pathogenic variants could finally achieve >99% sensitivity. However, there are also disadvantages of using next-generation sequencing as first tier screening test in NBS, such as the reporting of variants of uncertain significance and the high probability of detecting only one pathogenic or likely pathogenic variant with a high background of carrier rate. Hence, useful functional markers are still important.

## Figures and Tables

**Figure 1 IJNS-08-00064-f001:**
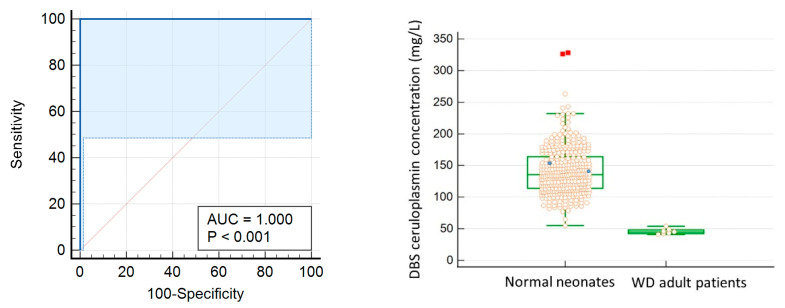
(**Left**) The ROC curve for diagnosis of WD using DBS ceruloplasmin level. To differentiate normal newborn from WD adult samples, the optimal cut-off value is 54 mg/L with 100% sensitivity and specificity. Area under curve is 1.000. (**Right**) Box-and-Whisker plot: solid circle: WD adult heterozygotes; red square: outliers.

**Table 1 IJNS-08-00064-t001:** Comparison of WD screening studies (Cp: ceruloplasmin; FP: false positive).

	Hahn et al. [4]	Ohura et al. [5]	Yamaguchi et al. [6]		Kroll et al. [12]	This Study
			(1993–95)	(1977–96)		
**Sample size**	3667	2789	126,810	24,165	353	359
**Age range**	3 m–15 y	1–6 y	Newborn	Late infancy to elementary school level	3 m–18 y	0–28 days and adult WD and carriers
**Analytical method**	Enzyme-linked immunosorbent assay	Enzyme-linked immunosorbent assay	Particle-coated fluorescence immunoassay		Enzyme-linked immunosorbent assay	Immunoturbidimetric assay
**Mean ± SD Cp level (mg/L)**	305 ± 95	124 ± 39.5	Unknown	Unknown	400 ± 144	141 ± 37.5
**Number of positive cases**	1 WD (32-month-old) with DBS Cp 23 mg/L	2 WD with neonatal DBS Cp 15 mg/L and 35 mg/L, respectively	953 FP No WD detected	5 WD Serum Cp < 100 mg/L	2 WD: neonatal Cp 26 mg/L and 28 mg/L, respectively	7 adults WD (41–54 mg/L)
**Remarks**	Repeated specimen was collected. Follow up for the positive case after second testing.	--	Different cut-offs High default rate (22% defaulted re-examination)	Consistent cut-off No default	--	--
**Conclusions**	Measurement of Cp level in DBS proposed as a reliable method for population screening of WD	CP level in DBS from children aged 1 to 6 years as a reliable marker for early detection of WD	--	Age of 3 years as the best point for WD mass screening	Presymptomatic screening for WD using DBS is possible, even in newborn	DBS Cp level measurement as potential marker for NBS WD

## Data Availability

The data presented in this study are available on request from the corresponding author. The data are not publicly available due to patient privacy issues.

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
