# Peer review of "Technical Study of Automated High-Throughput High-Sensitive Ceruloplasmin Assay on Dried Blood Spots—Reinstate the Potential Use for Newborn Screening of Wilson Disease"

_2409-515X, 2022, doi:10.3390/ijns8040064_

Round 1
Reviewer 1 Report (Previous Reviewer 1)
The present manuscript describes the adaptation of a technic to quantify ceruloplasmin in DBS, so it can be uses for Wilson disease NBS. This topic can be interesting for those working in the field. Nevertheless, there are some points that need to be reviewed:
- It is said that this screening test has 100% sensitivity and 100% specificity. Based on the results presented, this is a theoretical expectation not verified in practice. A pilot study was not made or samples from WD newborns tested. Also, there are some other disorders that can result in low ceruloplasmin levels, that may lead to false positive results, so the screening may not have 100% specificity. Please consider rewriting it (in the abstract and on lines 125 and 143)
- Although the use of adult WD samples could be used to verify the method, its direct comparison with NBS samples must be clearly indicated. Please consider indicating in figure 1 the origin of the samples.
Author Response
Thank you very much for the expert comments.
1) We rewrote the abstracts lines 20 to 24, "This test shows a high area under curve of 1.000, with 100% sensitivity and specificity in differentiating normal newborn from WD adult samples. However, the results should be further validated with true newborn WD patient samples together with the consideration of other factors which can also lead to low ceruloplasmin levels."
2) And rewrote lines 132 - 134 in Results, "This screening test shows a high area under curve of 1.000, with 100% sensitivity and specificity (Fig.1) in differentiating normal newborn from adult WD samples."
and on lines155 – 159, "At this cut-off, the test shows a high area under curve of 1.000, with 100% sensitivity and specificity in differentiating normal newborn from WD adult samples. However, the results should be further validated with true newborn WD patient samples together with the consideration of other factors which can also lead to low ceruloplasmin levels."
3) In Figure 1 legends, we specified the use of adult WD samples and normal newborn samples, as "Figure 1. (Left) The ROC curve for diagnosis of WD using DBS ceruloplasmin level. To differentiate normal newborn from WD adult samples, the optimal cut-off value is 54 mg/L with 100% sensitivity and specificity. Area under curve is 1.000. (Right) Box-and-Whisker plot: blue circle: WD heterozygotes; red square: outliers."

Reviewer 2 Report (New Reviewer)
An important issue and I am glad that the authors looked into it in details.
Major comment: the authors discuss the Roche method for ceruloplasmin. They discuss similar NBS project described by different authors but without any deep discussion on the method used for measuring ceruloplasmin. It is well recognised that enzymatic method is more accurate rather than immunoassay and it is worth elaborating on. Please see: DOI: 10.1177/0004563217695350
Author Response
Thank you very much for the expert comments.
1) We have added the ceruloplasmin methods of the discussed similar NBS project in Table 1.
2) We elaborate enzyme test on lines 58 – 59, "Ceruloplasmin activity has been shown to be both sensitive and specific for the detection of WD. But the assays are laborious and so far there is no commercial kit available."

This manuscript is a resubmission of an earlier submission. The following is a list of the peer review reports and author responses from that submission.
Round 1
Reviewer 1 Report
The present manuscript describes the adaptation of a technic to quantify ceruloplasmin in DBS, so it can be uses for Wilson disease NBS. This topic can be interesting for those working in the field. Nevertheless, the manuscript has some errors, incongruities and at several points lacks supportive information. Just to give some examples:
- Abstact - It is said that this screening test has 100% sensitivity and 100% specificity. Based on the results presented, this cannot be said. For example, no NBS samples from patients with WD were tested to address the effectiveness of the screening test.
- The modifications to the standard method were not highlighted and discussed.
- No data is presented to support the LOQ of 0.6 mg/L
- It is said that the best cut-off to differentiate normal newborns from WD newborns is 54mg/L, but no NBS samples from WD patients were tested.
- Ceruloplasmin values in normal newborns are compared with those from samples of older WD patients. This is not a good practice.
- On line 95 is presented a range for ceruloplasmin in DBS eluent from normal neonates and WD patients and them it is said that these ranges justified the need to decrease the LOQ to 0.6 mg/L. This is confusing! The units of the LOQ are by volume of blood (on the DBS) or by volume of eluate?
- On line 164 it is said that “Since WD is a genetic disease, the levels of ceruloplasmin of WD in neonatal period and adulthood should be comparable”. Although data is scarce, we can speculate that this can even be true. But what justifies it is not the fact of WD is a genetic disorder.
These are some examples of the reasons that make me not to recommend its publication.
Reviewer 2 Report
Summary
This paper describes a modification of the Roche/Hitachi Cobas c immunoturbidimetric human serum and plasma assay in order to measure ceruloplasmin in dried blood spots to identify newborns with Wilson Disease. It has potential in the future after further work.
General concept
The introduction needs to describe the current selection of methods for detection of patients with WD mentioning, early in the text, detection of low concentrations of ceruloplasmin in blood.
The reliability of each of the selection of tests for WD should be discussed in the introduction and why ceruloplasmin should be considered appropriate as a reliable newborn screening test, a method of choice.
Ideally neonatal dried bloodspots from known cases of WD should be retrieved from store to be included in the testing survey described. Indicate if this was not possible and if so why not.
If possible the recovery of ceruloplasin from stored bloodspots after weeks, months and years should be described. Is there a decline of measured concentration into the WD range from normal neonates?
Mention earlier in the text the known typical concentrations of ceruloplasmin in blood of adults versus newborns.
Materials and methods: Describe the usual Roche Cobas c502 method details and make it clear for the reader which details have been modified.
Briefly describe how the Roche Cobas works for the reader who is unfamiliar with the instrument. Are Cobas specific test tubes or microtitre plates loaded onto the machine? How many specimens can be loaded onto the machine in one batch? What do reagent 1 and reagent 2 comprise?
Describe how many analytical batches were run in this study and indicate whether the reagents were all from the same lot numbers. Report the within and between batch precision at or close to the diagnostic cut-off. Describe how many lot numbers of antibody were used in the study and whether the between batch precision changed with antibody lot number change.
Explain why ‘dried bloodspot’ calibrators were comprised of human plasma and not whole blood spotted onto collection paper.
Describe the punch and manufacturer that was used. Check that they were 3 mm and not 3.2 mm (quarter inch) diameter punches.
Describe the vessels the discs were punched into – test tubes, microtitre plates?
Describe the filter paper onto which dried bloodspots were collected and whether a minimum blood spot diameter was required.
Explain how the dried bloodspots were stored during the 2 years they were retrieved from.
Discussion.
This paper represents a preliminary technical investigation of measuring ceruloplasmin in dried bloodspots. The discussion is much longer than appropriate and should focus on the next steps for further trialling of the method to obtain a true sensitivity and specificity within the newborn population.
It is too early to state in Table 1 that DBS Cp level measurement is a reliable marker for NBS of WD. You do not yet have the evidence of reliability from neonates with WD.
Specific comments
Line 20: In the Abstract the sentence needs to make clear that the ‘true controls’ were from adult cases of Wilson Disease and not from neonates later confirmed to have WD. This should be clarified throughout the paper.
Line 38: confusing, I think you mean ‘in contrast’ not ‘on the contrary’.
Line 40: The reference Gao J, Brackley S, Mann JP . The global prevalence of Wilson disease from next-generation sequencing data. Genet Med 2019 May;21(5):1155-1163 might usefully be included here.
Line 68: f.a.s. PAC should be written in full before these abbreviations